# Radiofrequency Ablation, Cryoablation, and Microwave Ablation for the Treatment of Small Renal Masses: Efficacy and Complications

**DOI:** 10.3390/diagnostics13030388

**Published:** 2023-01-20

**Authors:** Lorenzo Bertolotti, Maria Vittoria Bazzocchi, Enrico Iemma, Francesco Pagnini, Francesco Ziglioli, Umberto Maestroni, Annalisa Patera, Matteo Pio Natale, Chiara Martini, Massimo De Filippo

**Affiliations:** 1Department of Medicine and Surgery, Section of Radiology, University of Parma, Maggiore Hospital, Via Gramsci 14, 43126 Parma, Italy; 2Department of Urology, Parma University Hospital, Via Gramsci 14, 43126 Parma, Italy; 3Department of Respiratory Disease, University of Foggia, Via Antonio Gramsci, 89, 71122 Foggia, Italy; 4Department of Medicine and Surgery, University of Parma, Maggiore Hospital, Via Gramsci 14, 43126 Parma, Italy

**Keywords:** small renal masses, ablation, imaging guide, renal cell carcinoma (RCC)

## Abstract

Over the last two decades the detection rate of small renal masses has increased, due to improving diagnostic accuracy, and nephron-sparing treatments have become the first-choice curative option for small renal masses. As a minimally invasive alternative, thermal ablation has increased in popularity, offering a good clinical outcome and low recurrence rate. Radiofrequency ablation, Cryoablation, and Microwave ablation are the main ablative techniques. All of them are mostly overlapping in term of cancer specific free survival and outcomes. These techniques require imaging study to assess lesions features and to plan the procedure: US, CT, and both of them together are the leading guidance alternatives. Imaging findings guide the interventional radiologist in assessing the risk of complication and possible residual disease after procedure. The purpose of this review is to compare different ablative modalities and different imaging guides, underlining the effectiveness, outcomes, and complications related to each of them, in order to assist the interventional radiologist in choosing the best option for the patient.

## 1. Introduction

Ablation of small renal masses was originally reported in 1997, and the first renal mass was treated in 1998. Since then, this treatment has gained increasing popularity [1,2].

The incidence of renal cell carcinoma (RCC) has grown during the last two decades owing to the higher detection rate of renal masses by ultrasonography (US), computed tomography (CT), and magnetic resonance (MR). RCC now accounts for 2% to 3% of all malignancies, with a greater frequency in Western nations. In years past, renal lesions were detected because they caused symptoms, but nowadays, renal masses are more often found incidentally and at a lower stage [3,4].

For a long time, the only curative option for RCC was radical nephrectomy. However, as surgical and imaging techniques advanced, nephron-sparing surgery became an option for smaller masses, and partial nephrectomy emerged as a second curative option that preserved renal function, especially in patients with comorbidities [5,6].

Ablation techniques might be nephron-sparing alternatives to partial nephrectomy and achieve the same oncological results. Indeed, in 2017, the American Urological Association approved ablative methods as valid alternatives to partial nephrectomy in pT1a renal masses (<4 cm). More recently, the EAU 2022 guidelines stated that ablative treatments are a viable option for certain patient categories; however, tumors larger than 3–4 cm and those located at the hilum or near the proximal ureter should not be treated with ablative therapies [7]. As a result, several investigations and research have been conducted on these techniques to expand their applicability [8,9]. 

Ablation requires pre-interventional imaging to examine all of a tumor’s characteristics; US, CT, or RM may be used. Typically, ultrasound and computed tomography (CT) are the most common procedures since the same imaging technology guides the procedure. Both CT and US have pros and cons to offer. Ultrasound provides a real-time image of the lesion, which makes it easier to make adjustments to the probe. On the other hand, CT guidance provides a more accurate analysis of the mass as well as any potential complications. The third alternative is a combined US–CT approach, which seems to be the optimal choice since it combines the best aspects of both. US may aid in detecting the tumor and guide the probe insertion in real time, whereas CT, with multiplanar reconstructions, validates the correct positioning of the probe and rules out any connection with surrounding organs.

Several modalities exist: Radiofrequency ablation (RFA), Cryoablation (CA), and Microwave ablation (MWA) are the most frequently described in the literature [10].

The purpose of this review is to compare different ablative modalities in terms of effectiveness, efficacy, complication, and oncological outcomes.

## 2. Patient Selection 

Patient selection is a crucial pre-requisite to guarantee an effective ablation. It requires a multidisciplinary meeting involving the interventional radiologist, the urologist, and the oncologist. 

Some laboratory tests should be run before the procedure, such as platelet count (should be >50,000/uL), INR ratio (should be <1.5), creatinine, and estimated glomerular filtration rate [11].

Since the treatment is generally conducted under sedation, a physical and pharmacological history should be acquired in order to check allergies or interactions with other medications. 

The ideal lesion has to be a RCC < 4 cm in diameter (pT1a), either exophytic or endophytic, in a patient who cannot undergo (or refuses) radical nephrectomy or general anesthesia. Patients usually present multiple comorbidities, advanced age, morbid obesity, or a solitary kidney [12,13]. Patients with several comorbidities or baseline renal impairment, as well as elderly patients, are especially exposed to the dangers of partial nephrectomy, such as ischemia time that might cause decreased renal function and bleeding. To reduce the risk of this complication associated with partial nephrectomy, they are often chosen for ablation techniques. 

Nevertheless, renal mass ablation can also be performed in candidates who underwent a previous partial nephrectomy or enucleation. It can be performed as well in the same kidney, even if the lesion has recurred after previous ablation.

The malignancy of the lesion must be proven either by pre-ablative biopsy that could be repeated in case of non-diagnostic result or through radiological features using standard criteria for solid lesions suspicious for aggressive behavior, including contrast enhancement and increasing diameter (>0.8 cm/year) during active surveillance, as it has been shown that one-dimensional growth rates are associated with tumor evolution [14,15].

The criterion for enrolling tumors into ablations should include not only biopsy-proven RCC but also imaging features [16]. In fact, even if up to 15–20% of small (<4 cm) masses that undergo resection are benign (e.g., lipid-poor angiomyolipoma and oncocytoma) [17], thus limiting the accuracy of tumor progression rate and recurrence, also pre-procedural biopsy are non-diagnostic in a percentage as high as 15–22% of cases [18], posing the problem of including or not in statistical analysis masses without a histological diagnosis that could still be malignant based on patient risk factor and radiological examinations [19].

## 3. Study of the Lesion—Imaging Guide

During ablation US, CT or MR can be used to target the lesion. The modality chosen is mostly determined by the operator’s preference or competence and by the availability of specific machinery, such as CT fluoroscopy or open MRI systems. Real-time fusion imaging combining ultrasound imaging with CT or MRI can also improve guidance and monitoring in tumor ablation procedures [1,20].

Ultrasound permits real-time positioning of ablation probes while avoiding ionizing radiation. The main disadvantage is that ultrasound remains extremely operator-dependent and could be problematic under certain circumstances, such as in people with large body habitus or when nearby intestinal gas obstructs the visibility of the tumor [21]. Some tumors may be difficult to visualize with US, especially as isoechoic lesions are not detectable unless the lesion protrudes beyond the normal renal margin. In addition, precise positioning in the planned point of the lesion could be difficult to reach and to assess, resulting in thinner safe ablation margins [22]. Image degradation happens during ultrasonography because of acoustic shadowing from the ice ball in CA and water vapor microbubbles in RFA, both of which appear hyperechoic and may restrict visibility and mask the target [23].

The advantages of CT over ultrasonography include being less operator-reliant, having no intestinal gas artifacts, and providing optimal views of the nearby critical structures. CT provides a better visualization of the lesion and the adjacent organs together with the precise positioning of the probe [22], and can easily guide ancillary but fundamental procedural tasks as hydro-dissection with the intervening colon when is needed. Moreover, using contrast-enhanced scans immediately after the procedure may draw attention to potential residual disease [23,24].

However, while lesion-visualization may be satisfactory in most cases, the use of a CT guide results in variable amounts of radiation exposure to the patient.

A traditional CT scanner or a cone beam CT scanner with real-time fluoroscopy can be used for percutaneous ablation, which may have consequences for dose reduction [25].

Another way to study and treat the lesion is with a combined approach, using both CT and US together. This strategy brings out the best aspects of both of them: it permits the operator to easily recognize the lesion and to direct the probe with the US dynamic live view, and then to complete the ablation accurately with the CT guidance. This allows multi-planar reconstructions after probe-positioning with a precise visualization of the tip and needles of the probe, simulating the ablation area and ensuring it to be in the planned position, or repositioning in cases where the probe is not exactly in the center of the tumor [26]. Andersson et al. documented a primary efficacy near 100% and a recurrence rate near 0% in a means follow-up of 2 ± 14 months, underlining that combined approach could be the best choice; however, it is just a preliminary result which has to be validated [26].

Moreover, PET/CT has been proposed as a viable alternative in some interventional treatments where lesions are difficult to see using traditional imaging, thanks to its advantage of combining anatomical and functional information [27,28].

MRI guide seems to have several advantages as it does not use ionizing radiation, provides good soft tissue resolution, and allows a live probe insertion. However, MRI is less frequently used for percutaneous ablation because of limited free scanning time and high cost, as percutaneous ablation by MRI guidance is conducted utilizing an open magnet, a standard solenoid magnet, or a specialized interventional magnet. Furthermore, the requirement for MRI-compatible ablation equipment is needed [27].

## 4. Ablation Techniques

There are different ablation modalities. The most performed in the literature are RFA, CA, and MW.

### 4.1. Radiofrequence Ablation (RFA)

Radiofrequency ablation (RFA), initially used to treat hepatic lesions, was once the most popular ablation technique for the treatment of RCC.

High-frequency alternating current (460–500 kHz) is used to administer RF ablation (RFA) through an RF electrode, which comes in a variety of sizes and forms. This creates ionic agitation, which results in frictional heat (temperatures of 60–100 °C); this process leads to the denaturation of cellular proteins and eventually to coagulative necrosis. Vaporization and eschara development at the electrode at temperatures exceeding 100 °C may reduce the effectiveness of ablation [29,30].

Using CT and US guide, the electrode enters the mass and the dedicated protocol tailored to the lesion is performed, including overlapping ablations in cases of tumors bigger than 3 cm.

Multi-tined systems improve heat delivery using RF both by increasing the cumulative surface area and by reducing impedance [31]. By allowing cooling fluid to circulate inside the electrode, temperatures at the electrode–tissue interface are reduced, decreasing charring and allowing for the increased deposition of power [32].

RFA is the most common ablation method because it is easily available, is cost-effective, and the ablation time is shorter compared to CA [9,33]. RFA has a higher success rate if the diameter of the lesion is <3 cm and if it is exophytic because it is easier to reach and treat it [18,29,33,34,35].

An important characteristic of RFA is the heat sink effect, generally known as a limit of this technique because the generated heat tends to decrease near structures with continuous liquid flow (large blood vessels ≥3 mm in diameter). This intrinsic feature can be useful to ablate perihilar tumors since it decreases the risk of damaging important anatomical structures with the high temperature. In addition, heat has an intrinsic coagulative effect, thus reducing the risk of minor hemorrhage [36]. Collecting system damage can occur as well, so it has been suggested placing a double J catheter in the ureter before the procedure in order to cool the collecting system (Figure 1). During the procedure, refrigerated water flows into the renal pelvis, protecting it from the heat [37].

Another disadvantage of RFA is heat dissipation from the top of the electrode, which can cause possible injuries of the surrounding structures [38].

### 4.2. Cryoablation (CA)

For Cryoablation, cryoprobes are usually connected to Argon, which can reach a temperature of about −190 °C in a very short time. The physical explanation is the Joule–Thomson effect. This involves the rapid expansion of a high-pressured gas through a valve. While the gas is insulated, a quick cooling effect is generated so that no heat is transferred to the surrounding environment. As a consequence of the quick cooling, this method is employed in CA to make an ice ball at the end of a cryoprobe.

The process of freezing and thawing damages the cell membrane and creates microvascular injury. This causes hypotonic stress, which leads cells to necrosis. 

Multiple probes can be used together at a minimum distance of 1–2 cm from one another. They can be placed forming various sizes and shapes. The probe can measure from 1.4 to 8 mm [39].

The optimal size of the created ice ball is 5 mm, with at least 3 mm exceeding the tumor so that the tissue can reach −20 °C [39,40].

Only using CA is it possible to create and visualize the ice-ball in real time using CT or MR, which defines the ablation zone. This is not possible by performing RFA or MWA. 

CA can allow the treatment of bigger lesions and, if diameter is >3 cm, oncological outcome is better compared to RFA [41]. Moreover, the injury rate near the collecting system for larger lesions is lower [9,33]. A further benefit of CA over heat-based ablation is that it is less painful, as cold itself acts similar to anesthetic.

On the other side, CA requires a longer time than RFA because of the freezing–thawing cycles, and the ice-ball does not have a uniform effect. Indeed, the very low temperature achieved inside the ice ball causes damage, while the zero degrees achieved on the perimeter of the lesion does not result in the death of cellular tissue [42]. 

As with RFA, CA can be impacted by nearby vascular systems and the resulting cold sink effect [43]. Indeed, Nonboe et al. showed in animal study that the blockage of arterial flow to the kidney increases the extent of the cryoablation zone by 80% [44]. CA is associated with a higher risk of hemorrhage because surrounding blood arteries are not directly cauterized, such as in RFA, because CA usually requires the insertion of multiple probes to create an adequate ice-ball, and because the procedure is longer, resulting in higher tissue damage.

### 4.3. Microwave Ablation (MWA)

An alternative ablative technique is MWA, which causes coagulative necrosis through electromagnetic waves reaching frequencies greater than 900 MHz [45]. An MWA probe causes water-molecule perturbation, leading to heat dissipation, then cellular death and tissue damage. 

MV ablation reaches a higher temperature over other heat-based techniques, which could be advantageous because it causes more intralesional damage and could help treating bigger lesions (diameter >3 cm) in a shorter amount of time [46]. Nevertheless, if the lesion is near the renal sinus, it could be dangerous because it can cause urine boiling and collecting system damage [47]. MW is less exposed to the heat sink effect, as there is no increase in tissue impedance, usually decreasing the deliverable energy. Yu et al. found in 2021 that 10-year survival results (such as overall survival OS, cancer specific survival CSS) were better with MWA over RFA; in particular, even the local tumor progression was significantly better with MWA (1.9%) than with RFA (8–9%) [48].

However, compared to FRA and CA, MWA could result in more pain for the patient [49].

For both heat-based and cold-based ablations, the imaging guide could be US, CT, or combined. Table 1 summarizes advantages and disadvantages of RFA, CA, and MWA.

Ablations can be performed under general anesthesia or local infiltration with monitored anesthesia care (MAC). Local infiltration is typically performed with an injection of 5–10 mL of 2% lidocaine at the skin, and the needle tract is used to access the lesion. In addition to local anesthesia at the site, sedation with hypnotics such as midazolam or droperidol and/or opioids such as fentanyl or meperidine can also be used to improve patient comfort for this procedure [50].

The efficacy of MAC with midazolam and fentanyl administration to general anesthesia for kidney PRFA was investigated1/20/2023 9:24:00 AM. It was found both better local tumor control and 3-year recurrence-free survival rates in patients treated under GA when compared to patients receiving MAC. These outcomes were probably related to better pain relief during general anesthesia, which resulted in the increased time of ablation treatment, with patients in the general anesthesia group undergoing a median treatment time of 25 vs. 16 min in those receiving MAC (*p* < 0.001). Additionally, the authors theorize that there is greater precision in targeting the tumor under general anesthesia as a result of controlled respiratory pauses [50]. Unfortunately, because many patients with these lesions have significant comorbidities, they may also not be ideal general anesthesia candidates and may have an increased risk of complications.

## 5. Outcomes

Clinical outcomes are invariably correlated with the size of the kidney tumor, regardless of the technique employed. 

RECIST criteria do not apply to evaluate the efficacy of ablative treatments for exophytic tumors.

In fact, the main indicator of necrosis in ablative procedures is not the reduction in the volume of the treated lesion but the absence of contrast medium impregnation in the various diagnostic investigations (CT, MRI, US) [51] (Table 2).

Another indicator of the efficacy of the percutaneous treatment of exophytic tumors is the frequent finding of a fibrous ring in the perirenal fat surrounding the lesion, the so-called “halo sign”, an expression of liponecrosis.

Adequately treated endophytic lesions do not soak in contrast medium and shrink in volume.

The technical success rate for ablations of T1a (≤4 cm) lesions is close to 100%, and the primary local control is excellent (Figure 2). The outcomes of T1a lesions treated with RF, CA, or MW ablation were comparable to those of total or partial nephrectomy (5-year cancer-specific survival 95–98%), but there were fewer post-procedural complications, unplanned hospital readmissions, decreased 30- and 90-day mortality, and a lower risk of long-term renal insufficiency [52].

Tumors bigger than 3 cm have been linked to a higher frequency of recurrence within the T1a classification, with a disease-free survival rate of 68% compared to 97% of tumors 3 cm or smaller [53].

### 5.1. Radiofrequency Ablation

Strong indicators of effectiveness include size (<3 cm) and exophytic tumor site. With recurrence rates as high as 14% in the T1b group, larger lesions have a worse success rate. Due to the heat sink effect of the surrounding bigger central arteries, central tumors are also less effectively treated.

### 5.2. Cryoablation

Patients who underwent partial nephrectomy, RFA or CRYO did not have any statistically significant differences in local recurrence, metastases, or death from RCC [54]. When compared to RFA, CRYO has been demonstrated to produce better oncologic results for tumors larger than 3–4 cm [41,55].

### 5.3. Microwave Ablation (MWA)

The short and intermediate results are similar to RFA and CRYO, with stated 3- and 5-year disease-free survival rates of 93% and 88%, respectively. Technical success and safety have been established [49,56]. Despite the larger median tumor size of 3.13 cm in the microwave group compared to 2.58 cm in the CRYO group, Jason Martin et al. found no difference in local or metastatic recurrence between the therapies for small renal tumors [57]. 

Shorter ablation and procedural times compared to other thermal ablation procedures are two benefits of MWA, along with less of a heat sink effect from the local blood supply than RFA and the capacity to produce bigger ablation zones [58,59].

## 6. Complications

To assess complications during or after renal mass ablation, Clavien–Dindo classification was used [60,61]. It consists of five grades (Table 3).

The most common complications after thermal ablation are grade I or grade II complications, according to Clavien–Dindo classification, thus they do not require surgical, endoscopical, or radiological intervention. Several minor complications could occur during or after the procedures, such as urothelial stricture and urine leak (more frequent when treating endophytic and perihilar masses), medical events (atrial fibrillation, hypertension, or supraventricular tachycardia), nerve injury, abscess, and pneumothorax (when the lesion is in the upper kidney pole, near the diaphragm). 

The most frequent major complications are hemorrhage, vascular injuries, and anemia. They are major complications with grade III of Clavien–Dindo classification, so they require surgical, radiological, or endoscopic intervention. 

Damage to nearby organs can occur as well: for example, if the renal mass is too close to the bowel, hydro-dissection (Figure 3) (for instance, injecting a glucose solution) could be a strategy to widen the space between the lesion and the organ and minimize possible lesions. 

For both heat-based and cold-based ablations, a higher complication rate is significantly related to greater tumor size and central location, according to Atwell et al. [55].

In general, the complications of thermal ablation, particularly regarding blood loss, need of blood transfusion, and shorter hospitalization, are lower over partial nephrectomy [62]. 

### 6.1. RFA

Weizner et al. described perirenal or retroperitoneal hematoma as the most common complication with an incident rate from 2% to 5%, and one case that needed embolization [63]. In general, the bleeding risk is lower compared to CA due to the cauterization effect, but the overall rate of complication between CA and RFA is statistically the same, as well as with the major complication rate [55]. Atwell et al., otherwise, described nerve injury as the most common complication, followed by urothelial stricture and then hemorrhage [55]. Nerve injury involves the genitofemoral or lateral femoral cutaneous nerve because the heat damages iliopsoas muscle. In all patients, symptoms were temporary and usually decreased after 6 months.

When treating endophytic lesions, calicopyelic damage can occur as well, but a ureteral double J catheter can be placed and refrigerating water can protect the collecting system during procedure [37]. 

### 6.2. CA

Atwell et al. described 7.4% of complications after CA (not higher than complications after RFA); two of them were major complications. Bleeding after ablation occurred in 4.8% of the patients with a retroperitoneal hematoma or hematuria, which represents the most frequent complication. Pulmonary embolus, infection, and medical events occurred in fewer patients [55]. Collecting system damage is rare in CA.

Shmidt et al. focused on CA in renal lesions with a diameter ≥3 cm and confirmed that its success rate is higher compared to RFA, but has an 8% of major complication rate (principally bleeding), and another 8% of minor complications occurred [41].

Some authors claim that damaging the renal collecting system with cold, rather than heat, is less likely to be associated with adverse consequences, allowing for potentially more aggressive treatment with cryoablation [64].

### 6.3. MWA

Yu et al. showed that the major complication rate (Clavien–Dindo grade III) was 3.7% and 3% (Clavien–Dindo grade IV), while Wells et al. and Moreland et al. described no severe complications, only Clavien–Dindo grade I or II [65,66]. MWA overall complication rate seems lower than that of CA. 

## 7. Current Guidelines

Current recommendations of the American Urological Association (AUA) advise that T1a renal masses (<3 cm) should be evaluated for renal thermal ablation; thus, partial nephrectomy remains the priority [67].

Thermal ablative therapies are acknowledged in the European Society of Medical Oncology (ESMO) guidelines as options for patients with small tumors (≤3 cm), particularly in patients who are frail, have a high surgical risk, have a solitary kidney, compromised renal function, hereditary RCC, or multiple bilateral tumors. Given this, the ESMO guidance emphasizes that the value of the existing evidence hinders solid judgments about the morbidity and oncological results of RFA and cryoablation [10]. 

The American Society of Clinical Oncology (ASCO) guidelines acknowledge the growing role of ablation, declaring that percutaneous thermal ablation should be evaluated for individuals with tumors for which complete ablation will be achieved, though partial nephrectomy is still recommended in patients whose tumors are suitable for it [68].

ASCO also emphasizes the degree of evidence supporting its recommendations as “intermediate-quality” and of “moderate” strength, in contrast to the AUA, which rates the quality of evidence comparing partial nephrectomy and ablation as “low”. ASCO reported one study that compared partial nephrectomy, RFA, and CRYO and found that the 3-year local recurrence-free survival for every modality was not statistically different (98% for all groups). As seen in previous studies, there was selection bias in the nephrectomy group of younger and healthier subjects, explaining the slightly better 3-year overall survival (OS) of 95% compared to 82% for RFA and 88% for CRYO [69].

The European Association of Urology published the guidelines update in 2022 declaring that thermal ablation is indicated in small RCCs in older patients with co-morbidities deemed unsuitable for surgical therapy, patients with a genetic proclivity for numerous tumors, as well as individuals with bilateral tumors or with a single kidney and a significant risk of full renal function loss after partial nephrectomy. It was additionally mentioned that due to the poor quality of the available data (level of evidence 3), no definitive conclusions could be drawn. Conclusions on morbidity and oncological outcomes for RFA and CA may be achieved in the future [3,70,71].

According to the 2017 National Comprehensive Cancer Network (NCCN) Clinical Practice Guidelines in Oncology, ablative procedures should be utilized only in eligible patients with clinical stage T1a RCCs because they are associated with a greater local recurrence rate than traditional surgery; in addition, there are only a few randomized phase III comparisons of ablative procedures with surgery. The guidelines also indicated that a biopsy could be contemplated in order to obtain or validate a diagnosis of malignancy and guide surveillance and ablation [72].

## 8. Conclusions

Percutaneous ablation of renal masses is a safe and effective nephron-sparing alternative. 

The target renal mass has a diameter <4 cm (T1a stage). RFA, CA, and MWA are effective treatment options in patients who cannot undergo a surgical intervention. In fact, in these selected patients, they lead to lower complication rates compared to partial nephrectomy, and a shorter hospitalization time. All three of them are safe and effective and if the mass is ≤3 cm in diameter, they show a technical success near 100% and a 5-year cancer-specific survival of about 95–98%. Complications can occur, but in the majority of cases they are rather modest (grade I and II according to Clavien–Dindo classification) [55]. RFA is suitable for both exophytic and endophytic lesions with a recurrence rate near 14%. CA is the best choice in patients with bigger tumors (>3 cm), with an overall complication rate similar to RFA. MW is a more locally aggressive ablation technique, more painful for the patient but with a lower local tumor progression [73]. 

Imaging guides play an important role in studying the lesion and treating it: the best outcome is reached using both US and CT, taking advantage of both imaging guidance modalities and limiting the patient’s exposure to radiations [23,26,74].

Since RFA, CA, and MWA share similar outcomes, once the imaging modality is picked, the choice between the three of them mostly depends on the operator expertise, which makes the difference in terms of outcome and complication rate.

## Figures and Tables

**Figure 1 diagnostics-13-00388-f001:**
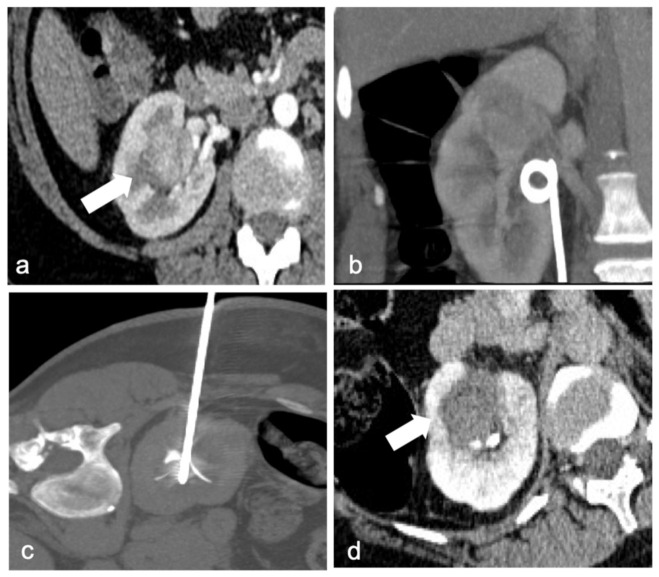
A 41-year-old woman with a biopsy-proven perihilar renal cell carcinoma of the right kidney ((**a**), arrow). Ureteral double-J stent was positioned right before the procedure to protect the pelvis and urinary tract with continuous refrigerated water flow (**b**). Precise positioning of the multi-tined RF probe in the right renal hilum mass (**c**). Contrast-enhanced CT after ablation shows the necrotic area (arrow), without any residual disease or immediate complication (**d**).

**Figure 2 diagnostics-13-00388-f002:**
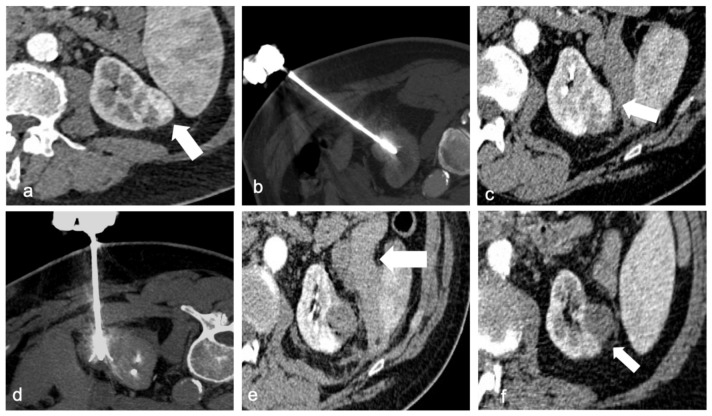
A 67-year-old man with a 34 mm renal cell carcinoma in the left kidney. Axial contrast-enhanced CT (**a**) demonstrates a heterogeneously enhancing mass (arrow). RF ablation was performed with multitined expandable electrods (**b**), but post-procedural contrast-enhanced CT revealed areas of enhancement in the mass (arrow) compatible with residual disease (**c**). A second target positioning was performed to achieve a complete ablation (**d**). Post-procedural CT shows no enhancement of the ablated lesion and a small hematoma (arrow, (**e**)). Follow-up CT after 4 weeks revealed no evidence of residual or recurrent disease (arrow) and partial reabsorption of the hematoma (**f**).

**Figure 3 diagnostics-13-00388-f003:**
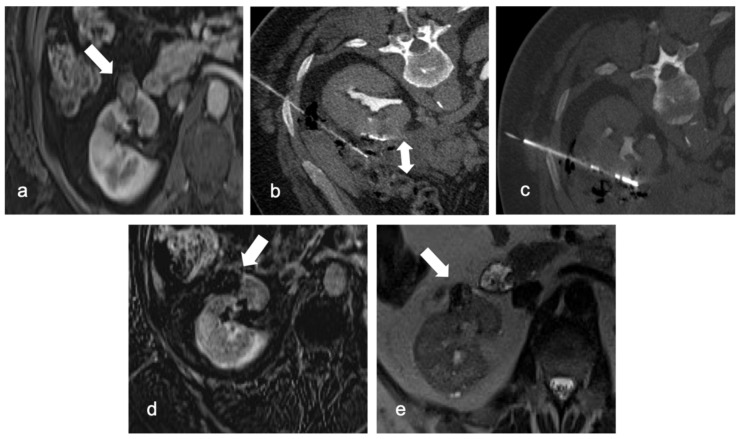
Arterial phase MRI shows a contrast enhancing anterior right kidney mass (arrow) in a 63-year-old man (**a**). Due to renal mass proximity to hepatic colon flexure, hydrodissection is obtained with 500 mL of glucose solution (double arrow, (**b**)). RFA is then performed with margin of safety between the probe and the colon (**c**). A follow-up MRI 2 months later revealed the necrotic mass (arrows) with no evidence of residual or recurrent tumor in the ablation zone: digital subtraction contrast-enhanced MRI (**d**) and T2w sequence (**e**).

**Table 1 diagnostics-13-00388-t001:** Advantages and disadvantages of RFA, CA, and MWA.

Ablation Technique	Advantages	Disadvantages
RFA (heat-based)	Lower bleeding rate Shorter procedural time than CA	Lower success rate if diameter of the lesion >3 cm No real-time visualization of ablation zone
CA (cold-based)	Real time visualization of ablation zone (ice-ball) Treat bigger lesions (>3 cm) Less painful than heat-based ablations	Requires longer procedural time Higher bleeding risk
MW (heat-based)	No heat sink effect Shorter procedural time than RFA and CA Treat bigger lesions (>3 cm)	More painful for the patient No real-time visualization of ablation zone More dangerous in hilar lesions

**Table 2 diagnostics-13-00388-t002:** Follow-up imaging after RFA: Appearance of a successful ablation.

	Immediate Post-Ablation	Follow-Up
CT	No evidence of contrast enhancement at CECT	Area of high attenuation (>40 HU) at nonenhanced CT ‘’Bull’s eye” or ‘’halo sign’’ appearance (exophytic lesion) Fat interposition between the ablated tissue and normal renal parenchyma (endophytic lesion) Distrophic calcifications
MRI	Loss of signal intensity on T2-weighted images	T2-weighted sequences: hypointense area relative to the normal renal parenchyma Totally dark on subtraction images

**Table 3 diagnostics-13-00388-t003:** Clavien–Dindo classification.

Grade I	Any deviation from the normal postoperative course without the need for pharmacological treatment or surgical, endoscopic, or radiological interventions Allowed therapeutic regimens are: drugs as antiemetics, antipyretics, analgetics, diuretics, and electrolytes and physiotherapy. This grade also includes wound infections opened at the bedside.
Grade II	Requiring pharmacological treatment other than such allowed for grade I complications. Blood transfusions and parenteral nutrition are also included.
Grade III	Requiring surgical, endoscopic, or radiological intervention
IIIa	Intervention not requiring general anesthesia
IIIb	Intervention under general anesthesia
Grade IV	Life threatening complication (including stroke and subarrachnoidal bleeding, but excluding TIA) requiring IC/ICU management
IVa	Single organ disfunction (including dialysis)
IVb	Multiorgan disfunction
Grade V	Death

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
