# Peer review of "Radiofrequency Ablation, Cryoablation, and Microwave Ablation for the Treatment of Small Renal Masses: Efficacy and Complications"

_diagnostics, 2023, doi:10.3390/diagnostics13030388_

Round 1
Reviewer 1 Report
Review Report
· In this article, the authors aimed to compare different ablative modalities of RCC, in terms of effectiveness, efficacy, complication and oncological outcomes
· The review is very interesting, written in a good fluent language, well-structured, and correctly organized. The authors have worked hard to present their opinion in a good manner.
· Although the topic is not novel, the article provides a comprehensive review of the different ablative modalities of RCC.
· I think it will contribute to the current literature.
· I have a few recommendations:
1. Add figures.
2. Tabulate some of your data.
Author Response
Dear Reviewer,
the response to the comment is in the attached file.
Thank you very much

Reviewer 2 Report
In the abstract: it is mentioned that ablation, should be offered to patients with comorbidities. This isn’t completely true nowadays.
Introduction:
What about the guidelines/recommendations of the EAU (admittedly indeed referred to in the ‘current guidelines’ section) ?
Why not performing lesional biopsy in all patients ? How can the ‘oncological outcome of ablation’ can be seriously interpreted when the histology of the treated lesion is unknown. Indeed, many small renal lesions are benign … Lipid-poor angiomyolipoma, oncocytoma; in addition differences in papillary RCC and the classical clear cell RCC …
Patient selection:
Is the lesional growth over time a reliable feature for decision of treatment ?
Study of the lesion:
Is ‘fusion’ of PET-CT images with CT (or US) during ablation feasible ?
Ablation technique:
RFA: is a needle with retrievable hook-umbrella preferable ?
Anesthesia ?
Outcomes:
See above comment in the introductory section.
Complication rates:
Does perprocedural lesional biopsy interfere with the complication rate ?
Radiologists’ awareness of imaging features after ablation … imaging can be rather tricky.
Author Response

(The authors gave the same response as above.)
